# Determining the depth of surface charging layer of single Prussian blue nanoparticles with pseudocapacitive behaviors

Ben Niu [1], Wenxuan Jiang[1], Bo Jiang[1], Mengqi Lv[1], Sa Wang[1] & Wei Wang [1✉]

Understanding the hybrid charge-storage mechanisms of pseudocapacitive nanomaterials holds promising keys to further improve the performance of energy storage devices. Based on the dependence of the light scattering intensity of single Prussian blue nanoparticles (PBNPs) on their oxidation state during sinusoidal potential modulation at varying frequencies, we present an electro-optical microscopic imaging approach to optically acquire the Faradaic electrochemical impedance spectroscopy (oEIS) of single PBNPs. Here we reveal typical pseudocapacitive behavior with hybrid charge-storage mechanisms depending on the modulation frequency. In the low-frequency range, the optical amplitude is inversely proportional to the square root of the frequency ($\Delta I \propto f^{-0.5}$; diffusion-limited process), while in the high-frequency range, it is inversely proportional to the frequency ($\Delta I \propto f^{-1}$; surface charging process). Because the geometry of single cuboid-shaped PBNPs can be precisely determined by scanning electron microscopy and atomic force microscopy, oEIS of single PBNPs allows the determination of the depth of the surface charging layer, revealing it to be ~2 unit cells regardless of the nanoparticle size.

[1] State Key Laboratory of Analytical Chemistry for Life Science, Chemistry and Biomedicine Innovation Center (ChemBIC), School of Chemistry and Chemical Engineering, Nanjing University, Nanjing 210023, China. ✉email: wei.wang@niu.edu.cn

Pseudocapacitors are electrochemical energy storage devices whose electrodes are made of redox materials that can undergo Faradaic reactions, while they exhibit surface-limited fast charging/discharging kinetics similar to those in capacitors[1–3]. Compared with ion batteries, pseudocapacitors are superior in terms of charging rate and power density because they are not limited by solid-state diffusion. At the same time, the energy density of pseudocapacitors is often several times higher than that of traditional electric double-layer capacitors, because the energy storage mechanism relies on redox reactions rather than simple electrostatic adsorption. Therefore, pseudocapacitors have received significant attention as they can establish a balance between power density and energy density.

In contrast to intrinsic pseudocapacitive materials such as $RuO_2$, $MnO_2$, and $Nb_2O_5$, whose pseudocapacitive features can be observed for not only nano-sized particles but also bulk materials, recent studies have demonstrated broad types of electroactive materials that exhibit extraordinary pseudocapacitive behavior only when the nanoparticle size is sufficiently small (several nanometers)[4]. These materials are referred to as extrinsic pseudocapacitive materials. Despite the debate on the validity of the concept of extrinsic pseudocapacitance[5,6], several potential candidates, such as $TiO_2$, $V_2O_5$, and metal-organic frameworks (MOFs), have emerged for the development of high-performance pseudocapacitors[1]. Hybrid charge-storage mechanism has often been proposed to understand the dependence of extrinsic pseudocapacitance on the morphology and size, including capacitor-like contributions from the surface or near-surface layer (surface charging) and battery-like contributions from the interior (diffusion-limited ion insertion). The two contributions can be distinguished by different electrochemical approaches such as cyclic voltammetry with varying scan rates, electrochemical impedance spectroscopy, and step potential electrochemical spectroscopy[7]. Although the surface charging layer that exhibits capacitor-like characteristics has been widely accepted in the literature[8,9], it remains technically challenging to precisely measure the depth of this layer, mainly because of the difficulty in determining the surface area of the electroactive materials in bulk electrodes.

The past decades have witnessed the emergence of single nanoparticle electro-optical imaging[10–13], which measures the electrochemical activity of single nanoparticles by monitoring its optical or spectroscopic signals under an advanced optical microscope as a function of the potential. Because the structures of the same individuals can be characterized using other techniques (e.g., electron microscopy), exploring the structure-activity relationship in a bottom-up manner is a promising perspective[14]. To date, different methodologies have been developed to study various electrochemical processes such as electro-catalysis[15,16], electro-deposition[17], electro-chemiluminescence[18], and batteries[19–21] at the single-nanoparticle level. For example, when applying a potential step to a single $WO_3$ nanoparticle and recording its optical signals as a function of time, Sambur et al. observed a sigmoidal-type kinetic curve that could be fitted with a model consisting of an exponential function, indicating surface-limited process in the initial stage, and a 0.5-order power function, indicating diffusion-limited process in the later stage[22–24]. Once the contribution from the surface layer was determined, it was possible to calculate the depth of the surface layer. Although powerful, this method relied on complex curve fitting of an optical chronoamperometric curve in the potential step, which comprised multiple ongoing processes of different timescales. Thus, it is desirable to separate the different contributions by choosing other electrochemical techniques.

Electrochemical impedance spectroscopy (EIS) is a frequency-resolved technique that is suitable for distinguishing different electrochemical processes based on frequency-dependent responses. It is implemented by applying a sinusoidal potential over a wide range of frequencies (typically kHz to mHz). The frequency modulation regulates the relative contributions of the resistance and capacitance elements to the total impedance, so that an equivalent circuit can be built to understand the complex processes during interfacial electron transfer[25]. For example, EIS is one of the most common and powerful techniques for exploring the energy storage mechanism of batteries, developing non-invasive chemical and biological sensors, investigating the corrosion of metals, etc.[26–28].

However, EIS of single nanoparticles is rare. Patrick et al. employed a scanning ion conductance microscope to measure the non-Faradaic EIS of single gold nanoplates[29]. The surface capacitance of single gold nanoplates can be mapped by scanning the electrochemical cell droplet over the surface. In contrast to the electrical readout, Tao and co-workers developed the plasmonic-based electrochemical impedance microscopy (p-EIM)[30–32]. In this method, the gold film acted as an optical-electrochemical conversion interface and exhibited a large background charging/discharging. Therefore, the optical amplitude was largest for bare gold electrode, and the presence of object (such as cell, bacteria, and nanomaterials) inhibited the background charging and decreased the optical amplitude. In other words, it was a 'turn-off' mode detection. In contrast, we employed a dark-field microscope (DFM) to measure the non-Faradaic EIS of single gold nanorods[33]. The principle is based on the dependence of the plasmonic scattering of gold nanorods on the electron density. The introduction of dark-field microscopy enabled a 'turn-on' version of optical impedance imaging which is more suitable for studying single nanoparticles. Faradaic EIS of single nanoparticles, however, has not yet been developed.

In this work, Faradaic EIS of single cuboid Prussian blue nanoparticles (PBNPs) was developed by imaging the periodic fluctuation amplitude of its resonant scattering intensity during sinusoidal potential modulations. The Faradaic reaction of PBNPs was selected as a model system for the following reasons. First, Prussian blue (PB) is a classical electrochromic material with well-documented crystal structures. The reduction of PB to Prussian white (PW) requires the collaborative uptake of one electron and one potassium ion ($K^+$), accompanied by an obvious color change from blue to colorless[34]. Single PBNPs exhibit significantly larger scattering cross sections for red light than PWNPs. Accordingly, when monitoring the scattering intensity of single PBNPs under a monochromatic DFM with red light illumination, the electrochemical reduction led to a dramatic decrease in the scattering intensity. If the electrode potential was sinusoidally modulated at a certain frequency, the scattering intensity of the single PBNPs fluctuated periodically at the same frequency. By extracting the optical amplitude and phase at a series of frequencies spanning four orders of magnitude, the optical electrochemical impedance spectroscopy (oEIS) can be performed at the single-nanoparticle level. Whereas the optical amplitude ($\Delta I$) was inversely proportional to the square root of frequency ($\Delta I \propto f^{-0.5}$) in the low-frequency range, it was inversely proportional to the frequency ($\Delta I \propto f^{-1}$) at a higher frequency. These results are consistent with the hybrid charge storage mechanism. Second, Prussian blue is a classical MOF material with well-defined pore structures. Owing to the high structural tunability, good electronic properties, and fast ionic conduction, MOFs are considered as one of the most promising candidates for developing high-performance pseudocapacitors[1]. The pseudocapacitive behaviors of both PB and PB analogs involve hybrid ion storage mechanism[35–38]. Despite the well-studied mechanisms, the depth of the surface charging layer remains unknown. Third, the PBNPs used in the present study have regular cuboidal geometry that can be well characterized by their length ($x$), width ($y$), and height ($z$) at the single-

nanoparticle level. Therefore, the depth of the surface charging layer can be determined as long as the contribution of the capacitor-like charging process is quantified through oEIS of the single PBNPs.

## Results

**Optical electrochemical impedance spectroscopy of single Prussian blue nanoparticles.** Cuboid PBNPs were synthesized according to a previously reported ultrasonic method[39] and were comprehensively characterized. The UV–Vis spectra showed a broad extinction band from 500 to 800 nm with a peak at ~700 nm (Supplementary Fig. 1a), owing to the intervalence charge transfer between Fe(III) and Fe(II) ions through the cyanide bridge. Scanning electron microscopy (SEM) and transmission electron microscopy (TEM) revealed a regular cuboid shape of these nanoparticles with sizes ranging from 50 nm to 1 μm (Supplementary Fig. 1b). It is well known that PB can exist in "soluble" and "insoluble" forms, which is related to its ability to remain in solution as a colloidal suspension[40]. The $K_{3p}$ photoelectron line in the X-ray photoelectron spectrum (XPS) revealed that the synthesized PBNPs were in the "soluble" form, with the chemical formula $KFe^{III}[Fe^{II}(CN)_6]$ (Supplementary Fig. 2). A couple of redox peaks in the cyclic voltammogram further indicated the excellent electrochemical activity of the PBNPs (Supplementary Fig. 1c).

The prepared PBNPs were deposited on an indium tin oxide (ITO)-coated glass slide and placed under a monochromatic DFM equipped with a light-emitting diode (LED, 660 ± 20 nm) as the light source (Fig. 1a). Because of the intensive absorption of red light, the resonant scattering of PBNPs resulted in high contrast in DFM for the visualization of single PBNPs as small as 100 nm (Supplementary Fig. 4). The electrochemical cell employed a two-electrode system[33,41]. Two pieces of identical ITO glass slides were placed face-to-face with a separation distance of 800 μm. PBNPs were immobilized onto the top electrode and served as the working electrode. Because the deposition density (i.e., surface coverage) of PBNPs was extremely low (~0.1%), the polarization at the two opposite electrodes was considered to be identical. In addition, the IR drop was negligible in 0.5 M $KNO_3$ electrolyte. Therefore, the potential of the working electrode was considered to be half of the total voltage generated by the potentiostat. We clarified that the electrochemical measurements were performed in a 0.5 M $KNO_3$ solution in the absence of additional redox molecules throughout the work. $KNO_3$ not only served as electrolyte to reduce IR drop, but also provided sufficiently high concentration of $K^+$ for insertion/extraction. Time-lapsed DFM images were captured by a monochromatic camera operating at 450 frames per second, allowing the simultaneous resolution of the scattering intensity curves of several tens of PBNPs. The data acquisition card recorded the transistor-transistor logic signal of the camera and the voltage of the potentiostat for precise synchronization between the electrical and optical signals. Specific details of the optical configurations are provided in Supplementary Information Section 2.

When a negative voltage is applied, the blue-colored PB is transformed into colorless PW, with an obvious change in the absorption spectrum. We then applied a linear descending voltage and observed that the scattering intensity gradually decreased from one plateau to another, indicating complete conversion from PB to PW (Fig. 1c). A formal potential of −25 mV was observed, indicating an approximate composition of 50% $KFe^{III}Fe^{II}(CN)_6$ and 50% $K_2Fe^{II}Fe^{II}(CN)_6$. Therefore, a sinusoidal modulation with an offset of −25 mV was applied in the EIS measurements (Supplementary Information Section 3). The scattering spectra of

a single PBNP were also recorded by applying different constant potentials at 5 mV (green), −25 mV (red), and −55 mV (blue) (Fig. 1d). This resulted in a clear dependence of the scattering intensity at 660 ± 20 nm on the potential.

We then explored the fluctuation of the scattering intensity along with a sinusoidal potential modulation (frequency 0.01 Hz, amplitude: 20 mV, offset: −25 mV). As shown in Fig. 1e, during the oxidation process (−45 to −5 mV), the scattering intensity gradually increased owing to the oxidation of PW to form PB. The intensity was subsequently recovered to its original value during the reduction process (−5 to −45 mV). The optical amplitude was defined as half of the peak-to-peak value in the scattering intensity curve, which was ~80 IU in this case. The optical phase was defined as the phase delay between the optical trajectory and potential curve, which was ~−7° in this case. To further demonstrate the cyclability, the optical amplitude of the single PBNPs was measured and was found to remain constant for 1000 consecutive cycles at a modulation frequency of 1 Hz (Fig. 1f). Additional details are provided in Supplementary Information Section 4. These results demonstrate not only the excellent cyclability of single PBNPs during consecutive potential modulations but also a unique strategy of using the scattering intensity of single PBNPs to quantify their redox state. These two features laid the foundation for the optical readout of the EIS of single PBNPs.

Next, the modulation frequency was adjusted from 0.01 to 100 Hz. The optical amplitude and phase were extracted and plotted as a function of frequency, thus generating the oEIS of the single PBNPs. The optical amplitude decreased with increasing frequency, spanning four orders of magnitude (Fig. 2a). Meanwhile, the optical phase changed from 0° to −90° (Fig. 2b). Further analysis revealed that while the optical amplitude was inversely proportional to the frequency ($\Delta I \propto f^{-1}$, Fig. 2c) in the high-frequency region (purple area, 1–100 Hz), it was inversely proportional to the square root of frequency ($\Delta I \propto f^{-0.5}$, Fig. 2d) in the low-frequency region (blue area, 0.04–0.8 Hz). These results are consistent with the pseudocapacitive behavior of PBNPs, which includes surface-limited capacitor-like behavior for high-frequency modulation and diffusion-limited battery-like behavior for low-frequency modulation. The oEIS of single PBNPs provides an opportunity to quantitatively distinguish the contribution from the two processes along the frequency dimension. Strictly speaking, it is more appropriate to describe the frequency-dependent optical amplitude/phase as an optical transfer function (OTF), instead of impedance. While the impedance can be calculated from OTF, we chose to display OTF in order to avoid the larger noise caused by the temporal derivative at higher frequency (see Supplementary Information Section 7 for details).

**Determining the depth of surface charging layer.** Because the optical intensity was quantitatively dependent on the oxidation state of the PBNPs, the optical amplitude indicated the amount of charge that was extracted or inserted into the single PBNPs during potential modulation (more experimental details are provided in Supplementary Information Section 8). At a high frequency, the surface charging mechanism was dominant. Rapid charge transfer at the Fe centers within the surface layer occurred through Faradaic redox reactions, exhibiting typical pseudocapacitive behavior, where the amount of charge was inversely proportional to the frequency (Fig. 2c). Higher the applied frequency, smaller the amount of charge transferred. When the frequency was low, PBNPs underwent a "shell-to-core" model[42], where the ionic diffusion in the interior regulated the charge transfer rate (Fig. 2g). Electrons and $K^+$ first filled in the surface

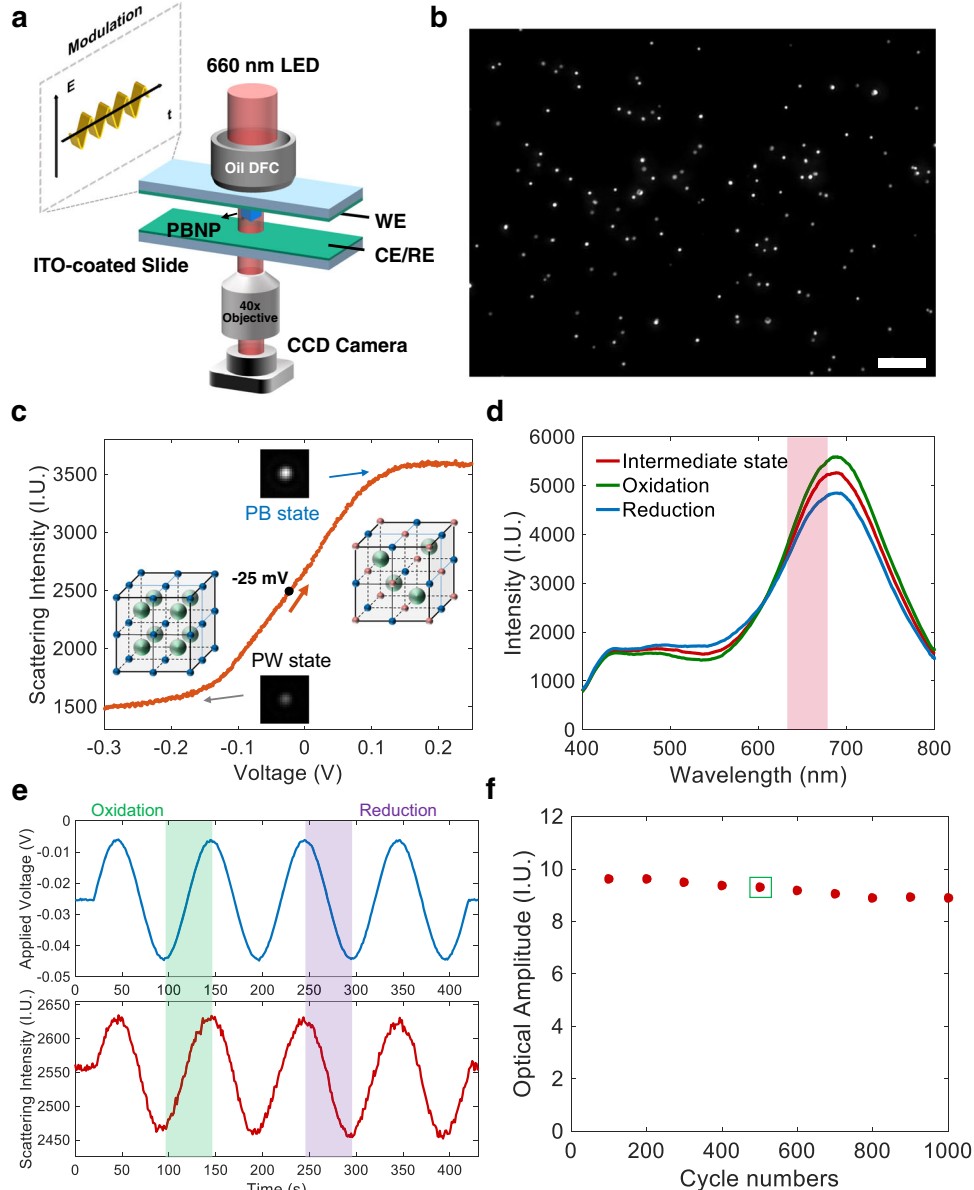

**Fig. 1 The dark-field device for extracting scattering intensity of single PBNPs during potential modulation. a** Schematic illustration of monochromatic DFM coupled with electrochemical cell to measure the oEIS of single PBNPs. **b** Representative dark-field image of several tens of PBNPs. Scale bar: 10 μm. **c** Clear dependence of the scattering intensity of single PBNPs with the electrode potential was observed; the scattering intensity of the oxidized state (PB) was 2.3 times higher than that of the reduced state (PW). Scan rate was 5 mV/s. Crystal structures of PB and PW are also shown; blue sphere: $Fe^{2+}$, red sphere: $Fe^{3+}$, green sphere: $K^+$. **d** Representative dark-field scattering spectra of single PBNPs at three different voltages: 5 mV (green), −25 mV (red), and −55 mV (blue). The red area is the wavelength range that we conduct dark-filed imaging. **e** Scattering intensity of single PBNPs as a function of sinusoidal potential modulation (frequency, 0.01 Hz; amplitude, 20 mV; offset, −25 mV). The green and purple area represent the oxidation and reduction process respectively. **f** Cyclability of single PBNPs for 1000 consecutive cycles during potential modulations (1 Hz); the average optical amplitude is extracted every 100 cycles. The detailed calculated process about the 400–500th cycles (green box inside) is provided in Supplementary Section 4. Source data are provided as a Source Data file.

layer, then diffused towards the inside of the nanoparticle. As a result, the amount of charge transfer was reversely proportional to the square root of the frequency as shown in Fig. 2d. When the frequency was lower than 0.04 Hz, the optical intensity reached a plateau and this area was called the depletion region in Fig. 2a (orange area, 0.01−0.025 Hz). It was because single PBNPs could only accept limited amount of charge depending on the volume. Therefore, the maximal optical amplitude at the plateau region was utilized to quantify the total charge ($Q_t$) that this particular PBNP could uptake.

When plotting the optical amplitude as a function of the inverse of the square root of the frequency ($f^{-0.5}$), it became clear that the curve was composed of two segments: a linear curve in the low frequency range (right part, $f^{-0.5}$), and a parabolic curve in the high frequency range (left part, $f^{-1}$ or $(f^{-0.5})^2$). It was well consistent with the proposed mechanism. The frequency that they intersected with each other was defined as the corner frequency, which represented the transition from capacitor-like to battery-like mechanisms. In order to unbiasedly determine the corner frequency ($f_{cutoff}$), a piecewise function (Fig. 2e inset, in which

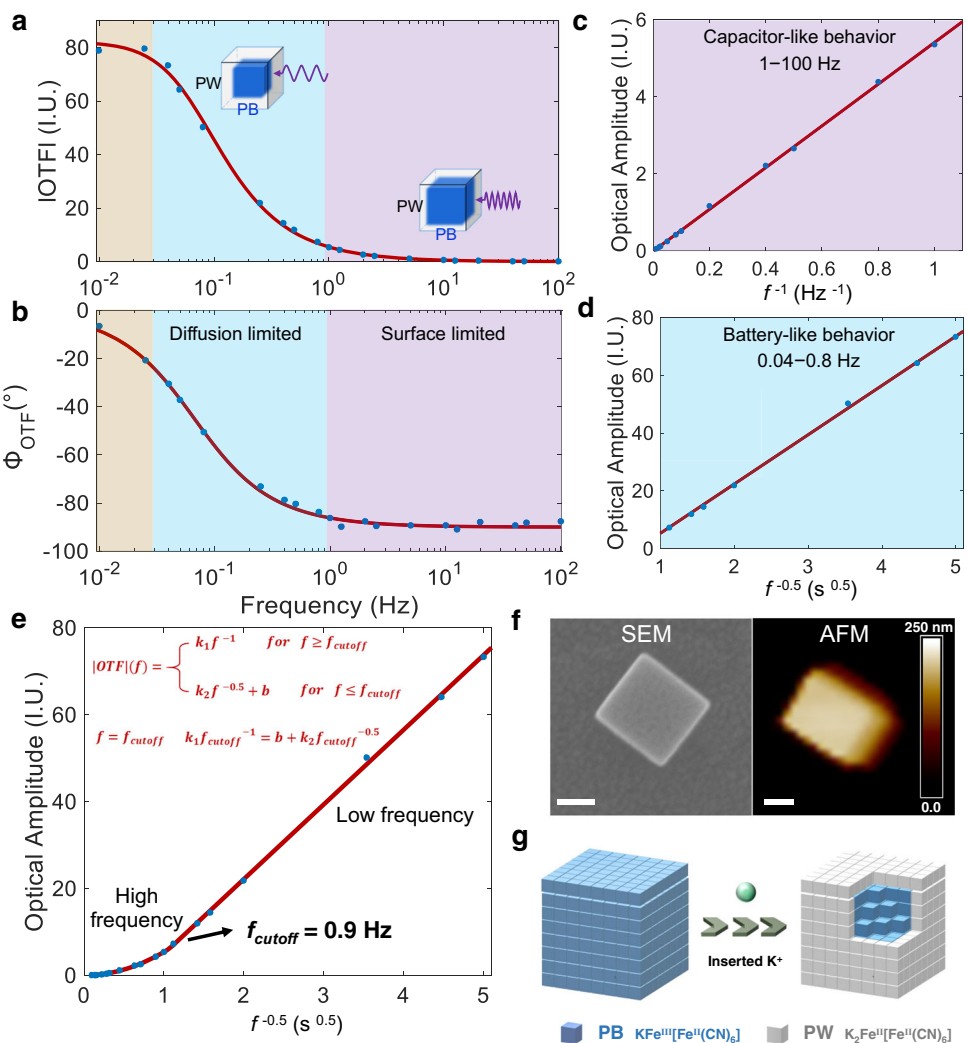

**Fig. 2 The oEIS of single PBNPs and corresponding mechanism analysis. a** Bode plots of OTF amplitude and (**b**) phase of single PBNPs as a function of modulation frequency. The purple, blue and orange area represent the high-frequency region, low-frequency region and depletion region, respectively. The red line shows the simulated results from equivalent circuit which will be discussed in the following section (function 3&4), and the blue dots are experimental data. **c** Surface pseudocapacitive behavior in the high-frequency region. **d** Diffusion-limited behavior in the low-frequency region. **e** When plotting optical amplitude as a function of $f^{-0.5}$, the dependence can be well fitted by a piecewise function as shown in the (**e**) inset to determine the corner frequency ($f_{cutoff}$). The red line is the fitted results, and the blue dots are experimental data. **f** Correlative SEM and AFM images of the same PBNPs examined in (**a–d**). Scale bar: 100 nm. **g** Schematic diagram of the "shell-to-core" model showing the transformation from entire PB to PW. Source data are provided as a Source Data file.

$f_{cutoff}$ is a parameter-to-be-fitted) was applied to fit the entire curve. For example, for the representative amplitude results shown in Fig. 2a, the corner frequency was fitted to be 0.9 Hz (Fig. 2e). Therefore, the contribution from surface charging ($Q_s$) was calculated to be 5.9 IU, accounting for 7.2% of $Q_t$.

We further examined the dependence of oEIS on the state-of-charge of single PBNPs by altering the offset potential. It was found that, while the maximal optical amplitude was indeed observed at the formal potential of PBNPs (−25 mV), the corner frequency and the depth of surface charging layer were independent on the state-of-charge (Supplementary Information Section 10), at least in the range of formal potential ±15 mV (corresponding to state-of-charge 30–70% of PB composition). It was attributed to the similarities in the crystal structure as well as lattice parameters between PB (oxidized form) and PW (reduced form).

Owing to the spatial resolution of DFM, after the electrochemical measurements, the same nanoparticle can be located and characterized by SEM and AFM to determine its three-

dimensional size. The length ($x$), width ($y$), and height ($z$) of the individual PBNP shown in Fig. 2a–d were determined to be 215, 200, and 215 nm, respectively (Fig. 2f). Geometrical calculations showed that the depth of the surface charging layer was 2.4 unit cells, corresponding to a volume fraction of 7.2% (see Supplementary Information Section 9 for details). In other words, it was the outer 2.4 unit cells of the single PBNPs that behaved like a capacitor ($f^{-1}$) and contributed 7.2% to the total charge storage capacity. Note that the actual depth could be smaller than 2.4 unit cells when considering the surface roughness of PBNPs. Given that the size of a unit cell of PB is 1.02 nm[34], the depth of the surface charging layer was determined to be 2.45 nm. The key to such calculations was the precise measurement of the surface area. Correlative oEIS and geometrical characterization ensured that the electrochemical activity and structural features were obtained for the same individual, underscoring the strength of single nanoparticle electrochemistry.

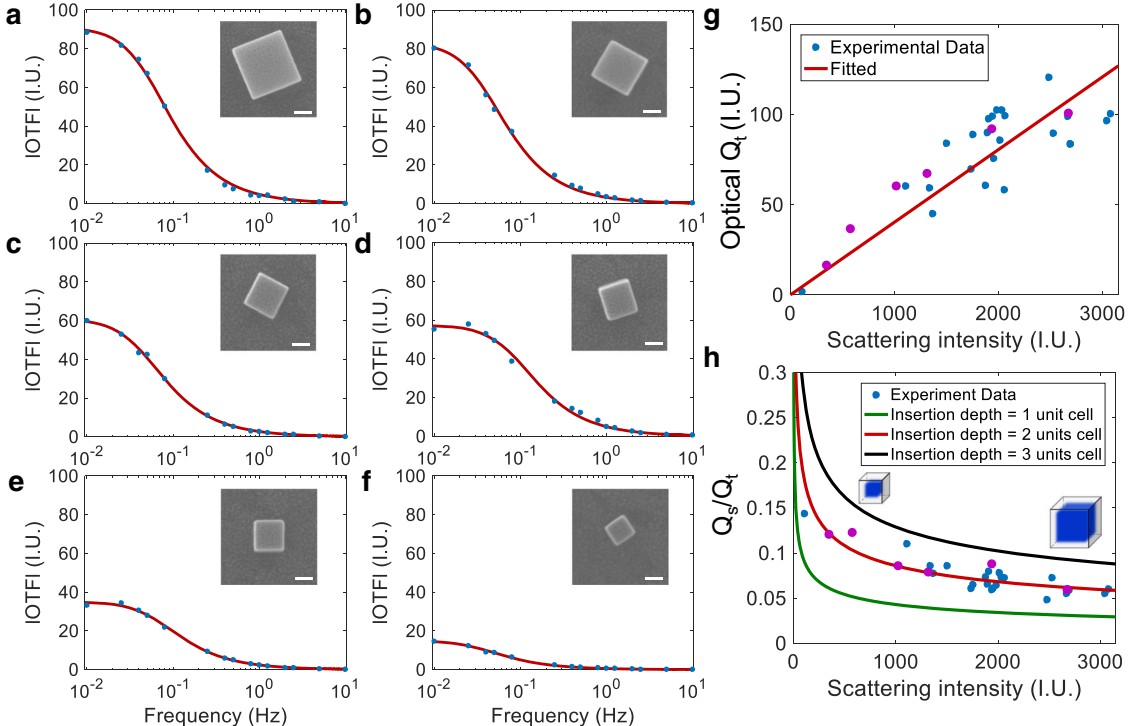

**Fig. 3 The oEIS of different single PBNPs individual and the corresponding statistical results. a–f** The oEIS of six representative single PBNPs. The red line shows the fitted results, and the blue dots are experimental data. The top right corner is the corresponding SEM image. Scale bar: 100 nm. **g** Dependence of optical $Q_t$ and (**h**) $Q_s/Q_t$ with the scattering intensity of single PBNPs by analyzing 30 individual species. Each dot represents an individual particle, and the dots corresponding to the six representative particles shown in (**a–f**) are marked in purple. The green, red, and black curves represent the theoretical $Q_s/Q_t$ curves as a function of the nanoparticle size by assuming a shell-layer depth of 1, 2, and 3 unit cells, respectively. Source data are provided as a Source Data file.

The wide-field feature of DFM also offered high throughput to perform the simultaneous oEIS and geometrical characterization of several tens of PBNPs in one experiment. Therefore, in this study, we examined 30 PBNPs with the formal potential of ~−25 mV and good morphology. The representative oEIS as well as the corresponding SEM images of the six PBNPs are shown in Fig. 3a–f. A sigmoidal dependence of the optical amplitude on frequency was observed for all the six PBNPs. They exhibited capacitor-like behavior in the high-frequency range and battery-like behavior in the low-frequency range (Supplementary Fig. 16). As shown in Fig. 3g, a linear correlation was observed between the maximal optical amplitude and the original scattering intensity; the latter is an indicator of its size (see Supplementary Information Section 13 for details). This was expected because the maximum optical amplitude was determined by the size, as discussed above.

More importantly, the contribution from surface charging gradually decreased from 14.4 to 5.5% when the nanoparticle size increased from 100 to 360 nm (Fig. 3h). This trend was also consistent with the assumption that the depth of the surface charging layer was constant regardless of the size of the nanoparticles. When the surface layer depth in a cuboid-shaped nanoparticle is one, two, or three unit cells, the theoretical volume percentages of the surface layer as a function of the size were calculated (Fig. 3h; green, red, and black curves, respectively). The experimental results can be best explained by assuming a surface layer depth of two unit cells. These results also suggest that smaller nanoparticles have a higher specific surface area and exhibit better electrochemical performance because more percentage of unit cells are exposed on the surface.

**Theoretical analysis of optical electrochemical impedance spectroscopy.** Theoretical analysis of the oEIS was conducted in two steps. First, the conventional (current-based) EIS of the entire ITO electrode was performed, and the spectrum was analyzed using a Randles-like circuit (Fig. 4a). Based on the above results acquired for all the PBNPs, theoretical analysis of the oEIS of the single pseudocapacitive nanoparticles was performed using the equivalent circuit model shown in Fig. 4b. Detailed descriptions on the elements ($R_s$, $C_{dl}$, $R_p$, $Z_w$, $R_{NP}$, $C_{NP}$) were provided in the Supplementary Information Section 11 and 15.

The EIS results for the entire ITO electrode are provided in the Supplementary Information Section 15. The solution resistance was determined to be 42 Ω, and the double-layer capacitance was 2.8 μF. Considering an electrode surface area of ~0.6 cm², the surface capacitance was 4.7 μF/cm², which is in good agreement with the previous values[33]. These results demonstrate the reliability of the electrochemical measurements in this study. More importantly, it is evidenced that nearly all the applied potential ($V_0$) dropped at the electrical double layer ($V_{dl}$) when the modulation frequency is lower than 100 Hz (Fig. 4b). Therefore, for the entire frequency range of the oEIS (0.01–100 Hz), the potential applied to the electrical double layer, and thus the nanoparticles within it, can be considered constant.

The equivalent circuit shown in Fig. 4b was employed to understand the oEIS of the single PBNPs, which consisted of a capacitor ($C_{NP}$) to demonstrate the charge-storage capability of PBNPs, and a resistor ($R_{NP}$) to demonstrate the charge transfer resistance and contact resistance[43] (Supplementary Information Section 10 and 11). While this equivalent circuit has been frequently adopted to explain the conventional EIS results of bulk pseudocapacitor electrodes[44], it is worth examining its suitability

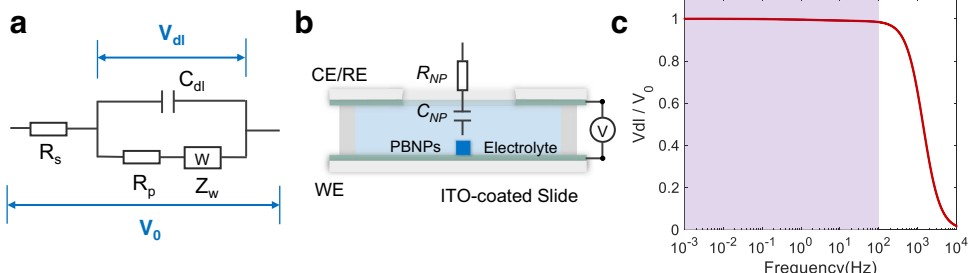

**Fig. 4 Theoretical analysis of the equivalent circuit. a** Randles-like circuit for the entire PBNPs-deposited ITO electrode. **b** Equivalent circuit consisting of a capacitor ($C_{NP}$) and a resistor ($R_{NP}$) to understand the oEIS of single PBNPs. **c** Within 0.01−100 Hz (purple region), nearly all the voltages from the potentiostat dropped at the electric double layer at the ITO-electrolyte interface, according to the conventional EIS analysis of the electrochemical cell.

for understanding the oEIS results of single nanoparticles. The Faradic impedance of a single PBNP can be expressed as [Eq. (1)]:

$$Z_F = R_{NP} + \frac{1}{j\omega C_{NP}}. \tag{1}$$

The current of a single PBNP can be simply expressed $i_F = V_0/Z_F$. As explained above, the optical intensity is proportional to the charge quantity $Q_F$, which can be derived by integrating the current. The amplitude and phase formula of $Q_F$ can be described as follows [Eq. (2), Eq. (3)] (see Supplementary Information Section 12 for details):

$$|Q_F| = \frac{V_0 C_{NP}}{\left(1 + \left(2\pi C_{NP} R_{NP} \cdot f\right)^2\right)^{1/2}}, \tag{2}$$

$$\Phi_{Q_F} = \tan^{-1}\left(\frac{1}{2\pi C_{NP} R_{NP} \cdot f}\right) - \frac{\pi}{2}. \tag{3}$$

Subsequently, the optical amplitude $\Delta I$ can be calculated from l$Q_F$l with a conversion factor $a$ [Eq. (4)]:

$$\Delta I = a \cdot |Q_F| = a \cdot \frac{V_0 C_{NP}}{\left(1 + \left(2\pi C_{NP} R_{NP} \cdot f\right)^2\right)^{1/2}}. \tag{4}$$

Here $a$ is the optical coefficient under specific experimental conditions and indicates the extent of change in the scattering intensity when one unit charge is inserted into the particle. Its value was experimentally determined to be $6.0 \times 10^{-5}$ IU/charge (see Supplementary Information Section 14 for details). Meanwhile, the phase of the optical impedance coincided with that of $Q_F$. This model can be well-fitted to the experimental oEIS curves (red solid curves in Figs. 2a, b and 3a–f).

When the frequency is high, $C_{NP}R_{NP}2\pi f$ is much greater than 1. A reasonable simplification of Eq. (2) leads to a linear dependence of the optical amplitude with the reciprocal of the modulation frequency [Eq. (5)]:

$$\Delta I = a \cdot \frac{V_0 C_{NP}}{\left(1 + \left(2\pi C_{NP} R_{NP} \cdot f\right)^2\right)^{1/2}} \approx \frac{aV_0}{2\pi R_{NP}} f^{-1}. \tag{5}$$

In the low-frequency range, diffusion plays a dominant role. The charge can be separated into two parts, and the Cottrell equation can be used as follows [Eq. (6)]:

$$\Delta I = aQ_0 + \int i_{dis} dt = aQ_0 + a \int kt^{-0.5} dt = aQ_0 + 2akf^{-0.5},$$

$$\text{where } k = \frac{nFAD_0^{1/2}C_0}{\pi^{1/2}}. \tag{6}$$

Here $Q_0$ is the charge related to the surface charging process, $n$ is the number of electrons involved in the redox reaction, $F$ is the

Faraday constant, $A$ is the surface area of the single PBNP, $D_0$ is the diffusion coefficient of $K^+$ in PBNP, and $C_0$ is the concentration of $K^+$ in the bulk solution. The excellent linear dependence of the optical amplitude with the square root of the reciprocal of modulation frequency (Fig. 2d) validates the applicability of Eq. (6). It also revealed the ionic diffusion coefficient to be $2.4 \times 10^{-12}$ cm$^2$ s$^{-1}$. This value is also consistent with a previously reported value of the potassium ion diffusion coefficient in PB nanomaterials[45].

## Discussion

In summary, we have employed a monochromatic DFM to optically acquire the Faradaic electrochemical impedance spectra of single PBNPs for the first time. Theoretical analysis was also conducted to understand the dependence of the optical amplitude and phase on the modulation frequency. oEIS of single PBNPs clearly revealed that the PBNPs exhibited hybrid ion storage mechanisms: pseudocapacitive behavior in the high-frequency region and diffusion-limited behavior in the low-frequency region. Piecewise function fitting of the optical amplitudes in high-frequency ($\Delta I \propto f^{-1}$) and low-frequency ($\Delta I \propto f^{-0.5}$) regions revealed the corner frequency to be 0.9 Hz. The contribution from surface charging ($Q_s/Q_t$) was 7.2% for a cuboidal PBNPs with a size of $215 \times 200 \times 215$ nm$^3$. Combined with oEIS of tens of PBNPs, we determined that the depth of the pseudocapacitive layer was ~2 unit cells. The present study not only proposes an unprecedented optical readout methodology for Faradaic EIS of single electroactive nanoparticles but also allows the determination of the depth of the surface charging layer through frequency analysis. Because Faradaic reactions often alter the dielectric constants (either absorption or scattering) of versatile electroactive materials, oEIS of single nanoparticles provides a bottom-up strategy to investigate the influence of particle geometry on impedance at the single nanoparticle level, with implications for understanding the structure-activity relationship of electroactive materials for use in energy storage devices.

## Methods

**Synthesis of Prussian blue nanoparticles**. The Prussian blue nanoparticles (PBNPs) were synthesized according to a reported ultrasonic method[39]. K$_4$[Fe(CN)$_6$]·3H$_2$O (1 mmol) powder was added into 0.1 mol/L hydrochloric acid aqueous solution. After immerging into water bath at 37 °C for 85 min under ultrasonic condition, the solution was cooled down to ambient temperature. The product was centrifuged at 14,800 rpm to collect the deposit. Finally, the obtained colloidal was washed with deionized water (DIW, 18 MΩ cm) and dried in the vacuum oven at 30 °C for further use.

**Electrochemical system and optical imaging device**. The electrochemical cell employed a two-electrode system[33,41]. Two pieces of identical ITO glass slides were placed face-to-face with a separation distance of 800 μm via Polydimethylsiloxane (PDMS). The thickness of ITO-coated glass slide was 1.1 mm (8 ohms/square). 150 μL droplet of five times diluted PBNPs solution was dropped on the ITO and dried in the vacuum oven for 12 h. The electrochemical measurements were

performed in a 0.5 M $KNO_3$ solution in the absence of additional redox molecules. Voltage was applied by the potentiostat (Autolab PGSTAT302N) and modulated via an external waveform function generator (RIGOL, DG1000Z). A data acquisition card (USB-6281, National Instruments) was utilized to synchronize the voltage from potentiostat and transistor-transistor logic signals from the camera.

The optical image of PBNPs was obtained with an inverted microscope (Eclipse Ti-U, Nikon), which was installed with an oil-immersed dark-field condenser (NA = 1.20−1.43), an objective lens (40×, NA = 0.6), and a 660 ± 20 nm light-emitting diode (M660L3-C1, Thorlabs) as the light source. The dark-field image was collected by a CCD camera (Stingray, Allied Vision Technologies).

## Data availability

The data that support the findings of this study are provided in the Source data file. Source data are provided with this paper.

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

## Acknowledgements

We thank the National Natural Science Foundation of China (W.W., Grants 21925403, W.W., Grants 21874070) and the Excellent Research Program of Nanjing University (W.W., Grant ZY JH004) for the financial support.

## Author contributions

B.N. and W.W. conceived the idea, designed the experiments, analyzed the data and wrote the draft together. B.N. carried out the experiments. W.J. helped to analyze the

data. B.J. helped to perform part of the additional experiment. M.L. helped in TEM and AFM characterizations. S.W. helped to synthesize the Prussian blue nanoparticles.

## Competing interests

The authors declare no competing interests.
