## [Peer Review File · Nature Communications]

REVIEWER COMMENTS

Reviewer #1 (Remarks to the Author):

Comments on the manuscript NComm-21-37657

Title: Determining the depth of surface charging layer of single Prussian Blue nanoparticles with pseudocapacitive behavior

Authors: Ben Niu, Wengxuan Jiang, Mengqi Lv, Sa Wang, Wei Wang

This manuscript introduces a methodology for calculating the size of the surface charging layer of nanostructured electroactive materials within pseudo-capacitive behavior. Particularly, the authors illustrate the method by using Prussian Blue (PB) nanoparticles as their experimental material proof-of-concept for inferring the size of the layer that is effectively related to the pseudo-capacitive response. The method comprises the usage of optical dark-field microscopy (DFM) output that is further correlated to the faradaic response (named as faradaic oEIS by the authors) of single PB nanoparticles. A cut-off frequency between two different behaviors of the optical intensities of the DFM output as a function of the frequency was identified (this needs clarifications – as detailed below). This cut-off (related to diffusive and non-diffusive regimes of the electrochemical charging load) was defined as the frequency value for inferring on the relative (in respect to the total) amount of charge that can be applied to calculate the fraction of the nanoparticle's surface with respect to the bulk, which would be responsible for an effective pseudo-capacitive contribution to the capacitive response of the nanoparticles. According to the author's premise, the charge state inferred from the cut-off obtained from their oEIS analysis corresponds to a fraction that can be directly correlated to the size of the nanoparticles aiming at to infer on the depth of the pseudo-capacitive layer. For the present experimental situation, the depth was defined as being around 2-unit cells of the PB structure, corresponding to a size of about 2 nm (which sound reasonable according to electrochemical theoretical methods).

Therefore, given the importance of this method for understanding and confirming theories that explain the nanoscale origin of the pseudo-capacitive behavior, which is a key capacitive behavior that impacts the development of supercapacitor and battery devices, my opinion is that the manuscript is worth of being considered for publication in Nature Communication. Nonetheless, the manuscript, is not fully conclusive yet. It needs major corrections and important clarifications before a final consideration of publication in Nature Communication can be taken. The reason for the needs for corrections and clarifications are stated below (and some are, in my opinion, mandatory):

1. The authors are not effectively measuring faradaic EIS using DFM (they called the response of DFM as oEIS). Technically, I understood that what is being measured is an optical transfer function – as a result of the time (or frequency) dependent perturbation of the potential to the optical time (or frequency) dependent response – that is proportional to the signal obtained in an original faradaic EIS, whose response to the potential perturbation is the electric (or electrochemical) current. I presume the optical density is proportional to the electric current in such a way that the admittance $[Y^* = 1/Z]$ $^* = \bar{i}/\bar{V}$ would be proportional to the optical transfer function such as $O^* = (\Delta I)/\bar{V}$, where ΔI is the optical intensity. Please, confirm and clarify if this is really what the authors meant. In this sense, can the authors demonstrate this more clearly in the SI document. Why only the modulus of O^* (referred as the amplitude of the signal) and phase were studied?

2. CRITICAL: I could not find anywhere the charge state of the nanoparticles with respect to their CV shown in the SI document (this was shortly explained – not with enough details – in the Figure 1 and lines from 139 to 151). Because this is quite important further detailed must be provided. This is a requirement because the pseudo-capacitance is proportional to the redox density-of-states which maximizes, for instance, in the Fermi (or formal) potential of the electrode (or at the nanoparticle). This must be clarified because it is key for the conclusions taken in the manuscript. If it is the case (which I doubt) that the state-of-charge would not control the size of the layer contributing to the pseudo-capacitive behavior in the nanoparticles, this must be demonstrated by the authors more clearly. In other words, if the useful layer responsible for converting Prussian Blue (PB) to Prussian White (PW) and vice-versa is independent of the state-of-charge and time of the electrochemical reaction this must be clearer. I presume this is not the case, but if it is the authors must make the correct arguments and demonstrations. In other words, to simplify my question, are the cut-off independent of the state-of-charge of the nanoparticles?

3. CRITICAL: What was the electrolyte used? There are some slight mentions to 0.5 M KNO₃ but this is not clear (as the pH is critical in the analysis of PB structures and state of charge this must be reported in more detail – experimental details of the manuscript must be improved).

4. CRITICAL: In the absence of more experimental detail on the type of electrolyte and electrochemical measurements (including the presence or absence of redox probe in solution – which complies or not with Figure 4) I am unable to evaluate the correctness of the equivalent circuit analysis conducted in section (theoretical analysis of oEIS) and equations employed therein. The theoretical analysis conducted in this section “theoretical analysis of oEIS” is straightforward, but to assume that the behavior of the oEIS measurement can be modeled by a RC circuit, it requires more arguments on the meaning of, at least, the capacitance (in the electrochemical context of this paper).

5. The key cut-off of the oEIS that defines the transition between diffusive and non-diffusive regimes at the frequency of 1.1 Hz are the core for this work. Can this cut-off be defined by plotting the optical amplitude as a function of the inverse of frequency and the inverse of the square root of the frequency in a unique superposed graphical for the readers taken their evaluation of this cut-off directly? The cut-off, as presented by the authors, is graphically undefined in the sense that the readers cannot take it from the graphical by themselves. Can the authors provide more details on how this are taken methodologically speaking?

6. Rigorously, methods based on frequency analysis requires validations generally using Kramers-Kronig methods applied on the frequency-dependent data. Do the authors consider this method as an important criterion of validation of time/frequency-dependent data (presumably on the $O^{\wedge} \text{function}$)? This would be important to validate the equilibrium condition for the charge state of the nanoparticles.

7. I agree with the equivalent circuit of (b) in Figure 4 – but this requires clarification on the meaning of the capacitance. In other works, this equivalent circuit can only be assumed under specific charge load of nanoparticles. It also depends on certain electrolyte conditions (in the absence of redox probe, for instance). The equivalent circuit of Figure 4(a) is not clear in which condition it was obtained (see my fourth comments above). There is a confusion here because the circuit of Figure 4a only applies for the case where there is redox probe in solution (the traditional Randle circuit) and in which the Warburg element of the circuit represents the diffusion of ionic species. This circumstance is quite different from the non-diffusive case. Please, can the authors clarify? It is important to reconcile between diffusion and non-diffusive regime by addressing a correct meaning for the capacitance of the nanoparticles under the two regimes of reactions. The meaning of an electrochemical capacitance can be applied in both cases without the need of a Warburg element. Otherwise, there is a mixture of two analysis that are quite different in essence and that cannot be modelled as the authors did.

8. Only with the further clarifications demanded in my comment number 7 is that I will be able to adequately validate the analysis conducted by the authors in equations (1) to (6).

Reviewer #2 (Remarks to the Author):

The authors present an electro-optical imaging approach that measures the scattered light intensity of single Prussian blue particles as a function of an applied voltage waveform. One major advance of

this study over previous single nanoparticle electro-optical studies (refs 22-24) is the electrochemical impedance waveform, which provides unique insight into the ion insertion process. The scattering intensity versus frequency supports two different ion insertion kinetic regimes (i.e., mass transport-limited regime and surface kinetic-limited regime). This paper is a very nice piece of work and should be published in Nature Communications. However, it would be helpful to the community if the authors could clearly explain how the scattering signal quantitatively relates to charge inserted in the particles (see comments below).

1. Can the authors clarify what they mean by “an equilibrium potential of -25 mV was observed” in line 133.

2. In lines 130-134, the authors begin to imply that the scattering signal is immediately proportional to charge inserted in the particle. It might be useful to explain at this point that the scattering signal does not immediately report on the ion insertion content in the host. In line 181, the authors state “the optical intensity was quantitatively dependent on the oxidation state of the PBNPs”. At this point in the manuscript, the authors have only demonstrated that the scattering signal changes in a rational way with the applied potential waveform. There is no quantitative relationship between charge inserted in the particle and the scattering signal (yet).

3. In line 193-194, the authors assume “the maximal optical amplitude at the plateau region was utilized to quantify the total charge that the particular PBNP could uptake”. A very recent Nature paper by Merryweather et al. also used a scattering method to study Li-ion insertion in LiCoO₂ particles. It is not straightforward to relate the scattering intensity to ion insertion content. Can the authors provide more justification for this important assumption in this work (Merryweather, A. J.; Schnedermann, C.; Jacquet, Q.; Grey, C. P.; Rao, A. Operando Optical Tracking of Single-Particle Ion Dynamics in Batteries. *Nature* 2021, 594 (7864), 522–528). Can the authors consider details in the SI of that Nature paper and explain how the proposed method in Section 6 of the SI is a reliable approach to quantitatively relate scattering intensity to charge inserted?

4. What do the authors mean by “corner frequency” in line 196?

5. In line 213, the authors state they examined 30 PBNPs with “good activity”. There are more than 30 eligible objects in Fig 1b. Does this mean not all particles are active? If so, then the above assumption in comment 3 deserves more serious attention because the applied potential is applied to the entire electrode but the amount of charge injected into every particle is different. The “proportionality constant” approach described in Supplementary Section 6 does not explain the fraction of inactive particles.

6. In figure 2b, the authors could consider using mass transport-limited or diffusion-limited and surface-limited instead of diffusion-control and pseudocapacitance-control.

Reviewer #3 (Remarks to the Author):

This is a very interesting work that applies the methodology of electrochemical impedance spectroscopy (EIS) at the single nanoparticle level. It is performed on PB prussian blue nanocubes. This choice of nanomaterial is relevant because it can be considered as a model of a solid-state charge storage nanodevice. While most charge storage activities of nanomaterials are studied at the level of a large ensemble, here the charge storage performance is studied at the level of a single particle, through the indirect study of the variation of its optical properties during EIS solicitation. The amplitude and phase of the optical response of individual nanocubes to a sinusoidal potential waveform are then discussed and analyzed as would be the response to electrochemical current.

Various relevant and innovative information is obtained from this work. It totally fulfills the standards of Nature Communications. However the following points should be discussed or reconsidered before publication.

1. First of all, the methodology is sound and innovative. Indeed, works proposing an EIS analysis at the level of the individual NP are rare. It is also the first time that it is applied to the monitoring of individual nanoparticles by dark field optical microscopy. In this respect it paves the way to many other capacitance and impedance assessment at the single NP level. However, it seems that the concept of optical impedance spectroscopy has been quite developed and implemented in SPR microscopy by the late NJ Tao's group (even with the help of the lead author of this work). This should be recalled in the context and state of the art of this SPR. A comparison of the two optical EIS implementations or methodologies, e.g., in terms of sensitivities or temporal/frequency resolutions, could also be appreciated.

2. The context of the study is related to pseudocapacitance. The authors recall the controversy central to this notion, not always accepted, where both double layer and faradaic behavior are observed. However it is quite difficult to apprehend from the data presented whether the system response belongs or not to a pseudocapacitance behavior.

2.1 Indeed the EIS response is modelled by a serie RC circuit which may be an oversimplified description of a pseudocapacitor. Even ref 38 states that the RC serie description is too simple and a transmission line behavior would be more appropriate. It is indeed expected that the EIS response would show some charge transfer processes with diffusion and capacitive contributions. See for

example the guidelines suggested by Gogotsi and Simon Adv Energy Mater 2019 1902007. It is then possible that the EIS data analysis is over interpreted as yielding pseudocapacitance behavior. Plotting the data in a Nyquist plot would show the EIS shows only a RC type semi-circle response without diffusive or even pure capacitive branches expected for the pseudocapacitor systems. One could then wonder if the system can be really named pseudocapacitor. It may be that the NP behavior is very different from the generally studied film electrodes.

2.2 One way to confirm the behavior would be to perform the same analysis at another state of charge of the NP, say another electrode potential, especially since there is a large potential range which can be explored without going to full charge/discharge of the material.

2.3. If the trend is not confirmed at other NP states of charge I wonder if the regimes presented in Fig 2a and 2b are correct. The pseudocapacitance region seems a pure capacitance behavior to me. The diffusion regime is rather a RC behavior while at lower frequency a pure resistive behavior is observed.

2.4. Definitely, the conclusion relative to the thin depth of charging is likely correct. Indeed the apparent optical 'resistance' corresponds to a very small amount of charge/discharge when comparing the ca.80 optical units of the 25mV amplitude pulse compared to the 2000 optical units required for full charge/discharge (Fig 1). As a comment, this ensure that the EIS measurement is indeed not perturbing much the overall composition of the NP, as requested for EIS experiments.

3. The open circuit charge state of the NP is approximately 0.5. Is this value common or was it intentionally obtained during synthesis? This should be detailed.

4. Actually it also means that the NP studied by EIS is half-charged and that the scheme in fig 2f presented to describe the PB charge/discharge is not correct. The calculation of the depth of surface charging may also be underestimated.

5. The same authors have published other works related to the same PB NPs with dark field monitoring. One of their conclusions was that the NP activity depended strongly on the contact resistance to the electrode material. I wonder how they could circumvent this issue in the present work? I also wonder if finally this limitation is not the charge transfer resistance revealed by the EIS evaluation?

6. As a general comment, pseudocapacitors have been mostly studied by cyclic voltammetry, for example by varying the CV scan rate. This would be equivalent in some respects to the EIS measurement. Did the authors considered such studies which would more visually present the pseudocapacitor behavior?

7. Finally, one may have shown the electrochemical behavior of large ensemble or film of NPs showing for example their electrochemical current based EIS rather than the oEIS. One could wonder if the oEIS behavior fits the electrochemical EIS one?

Responses to Reviewers' Comments

Reviewer #1 (Remarks to the Author):

Comments on the manuscript NComm-21-37657

Title: Determining the depth of surface charging layer of single Prussian Blue nanoparticles with pseudocapacitive behavior

This manuscript introduces a methodology for calculating the size of the surface charging layer of nanostructured electroactive materials within pseudo-capacitive behavior. Particularly, the authors illustrate the method by using Prussian Blue (PB) nanoparticles as their experimental material proof-of-concept for inferring the size of the layer that is effectively related to the pseudo-capacitive response. The method comprises the usage of optical dark-field microscopy (DFM) output that is further correlated to the faradaic response (named as faradaic oEIS by the authors) of single PB nanoparticles. A cut-off frequency between two different behaviors of the optical intensities of the DFM output as a function of the frequency was identified (this needs clarifications – as detailed below). This cut-off (related to diffusive and non-diffusive regimes of the electrochemical charging load) was defined as the frequency value for inferring on the relative (in respect to the total) amount of charge that can be applied to calculate the fraction of the nanoparticle's surface with respect to the bulk, which would be responsible for an effective pseudo-capacitive contribution to the capacitive response of the nanoparticles. According to the author's premise, the charge state inferred from the cut-off obtained from their oEIS analysis corresponds to a fraction that can be directly correlated to the size of the nanoparticles aiming at to infer on the depth of the pseudo-capacitive layer. For the present experimental situation, the depth was defined as being around 2-unit cells of the PB structure, corresponding to a size of about 2 nm (which sound reasonable according to electrochemical theoretical methods).

Therefore, given the importance of this method for understanding and confirming theories that explain the nanoscale origin of the pseudo-capacitive behavior, which is a key capacitive behavior that impacts the development of supercapacitor and battery devices, my opinion is that the manuscript is worth of being considered for publication in Nature Communication. Nonetheless, the manuscript, is not fully conclusive yet. It needs major corrections and important clarifications before a final consideration of publication in Nature Communication can be taken. The reason for the needs for corrections and clarifications are stated below (and some are, in my opinion, mandatory):

Response: We very much appreciate this reviewer's valuable comments that have helped us to significantly improve the manuscript. Detailed point-by-point responses are provided below.

1. The authors are not effectively measuring faradaic EIS using DFM (they called the

response of DFM as oEIS). Technically, I understood that what is being measured is an optical transfer function – as a result of the time (or frequency) dependent perturbation of the potential to the optical time (or frequency) dependent response – that is proportional to the signal obtained in an original faradaic EIS, whose response to the potential perturbation is the electric (or electrochemical) current. I presume the optical density is proportional to the electric current in such a way that the admittance $[[Y^*=1/Z]] \hat{=} \tilde{I}/\tilde{V}$ would be proportional to the optical transfer function such as $O^*=(\Delta I)/\tilde{V}$, where ΔI is the optical intensity. Please, confirm and clarify if this is really what the authors meant. In this sense, can the authors demonstrate this more clearly in the SI document. Why only the modulus of O^* (referred as the amplitude of the signal) and phase were studied?

Response: We agree with this reviewer that, it is more appropriate to describe the frequency-dependent optical amplitude/phase as an optical transfer function (OTF), instead of impedance/admittance.

We chose to display the OTF (rather than impedance directly) because of the following reasons. In order to obtain the impedance/admittance information, a first-order derivative had to be performed to the optical intensity curves (corresponding to charge quantity) to resolve the current (corresponding to charge transfer rate, please refer to our previous work *J Am Chem Soc*, 2017, 139, 186 for details). Unfortunately, the first-order derivative was found to significantly increase the noise level, particularly at high-frequency range (Fig. R1). It made the quantification at high frequency region challenging because the signal itself became smaller when the frequency was higher.

Fig. R1 Left panel: (a) Representative scattering intensity curve under a modulation frequency of 1 Hz and (b) its first order derivative. The corresponding Fourier transform results are shown in the right panel.

Although it was difficult to directly measure the current at each frequency, we would like to kindly point out that, one was able to build a mathematic transform to obtain the impedance/admittance (Z , voltage ~ current) from OTF (charge quantity or the integration of current ~ voltage):

$$|Z| \propto \frac{1}{f \cdot |\text{OTF}|}, \quad \phi_Z = -\phi_{\text{OTF}} - \frac{\pi}{2}.$$

By doing so, the amplitude and phase of optical impedance can be calculated from the original OTF data (as shown in Fig. R2).

Fig. R2 The amplitude ($|Z|$, left panel) and phase (ϕ_Z) of impedance can be calculated from those of OTF. The red lines are fitted results and the blue dots are experimental data.

According to this reviewer's comments, we have revised the manuscript (Manuscript Page 6) and the Supplementary Information (Section 7) to better clarify this point.

2. CRITICAL: I could not find anywhere the charge state of the nanoparticles with respect to their CV shown in the SI document (this was shortly explained – not with enough details – in the Fig. 1 and lines from 139 to 151). Because this is quite important further detailed must be provided. This is a requirement because the pseudo-capacitance is proportional to the redox density-of-states which maximizes, for instance, in the Fermi (or formal) potential of the electrode (or at the nanoparticle). This must be clarified because it is key for the conclusions taken in the manuscript. If it is the case (which I doubt) that the state-of-charge would not control the size of the layer contributing to the pseudo-capacitive behavior in the nanoparticles, this must be demonstrated by the authors more clearly. In other words, if the useful layer responsible for converting Prussian Blue (PB) to Prussian White (PW) and vice-versa is independent of the state-of-charge and time of the electrochemical reaction this must be clearer. I presume this is not the case, but if it is the authors must make the correct arguments and demonstrations. In other words, to simplify my question, are the cut-off independent of the state-of-charge of the nanoparticles?

Response: We agree with this reviewer that the pseudocapacitive behavior was significantly dependent on the state-of-charge of redox nanomaterials. According to this reviewer's comments, we had conducted further experiments at different offset potentials to examine the influence of state-of-charge on the OTF as well as corner frequency. It was

found that, while the maximal optical amplitude was indeed observed at the formal potential of PBNPs (-25 mV), the corner frequency and the depth of surface charging layer were more or less independent on the state-of-charge, at least in the range of formal potential ± 15 mV (corresponding to state-of-charge 30~70%).

Fig. R3 (a) The dependence of the scattering intensity of single PBNPs on the potential. Scan rate is 5 mV/s. Representative scattering intensity curves are provided when the modulation frequency is 0.01 Hz with the same amplitude of 20 mV, and corresponding offset potentials are -10 mV (b), -25 mV (c), and -40 mV (d), respectively.

First, optical response as a function of sweeping potential from -300 to +250 mV (scan rate: 5 mV/s) was provided in Fig. R3a. It revealed a formal potential of -25 mV, and a quasi-linear dependence of optical intensity on potential during -55 and 5 mV.

Then, oEIS of the same individual PBNP was measured at varying offset potentials of -10, -25, -40 mV with the same amplitude of 20 mV. The results are shown in Fig. R4. According to the curve shown in Fig. R3a, the state-of-charge under these potentials was 70% PB-30% PW, 50% PB-50% PW, 30% PB-70% PW, respectively. It was clear that the maximal optical amplitude was obtained at the formal potential of -25 mV. However, similar corner frequencies (0.7 Hz, 0.8 Hz and 0.8 Hz) and percentage (7.1%, 7.6% and 7.4%) were observed in different offset potentials (-10, -25 and -40 mV). We attributed such stability as the same face-centered cubic crystal structure and the similar lattice parameters between Prussian Blue (oxidized form) and Prussian White (reduced form).

Fig. R4 (a) The OTF amplitude of the same individual PBNP at varying offset potentials of -10 mV, -25 mV and -40 mV. Surface-limited behaviour in the high frequency region (left panel) and diffusion-limited behaviour in the low frequency region (middle panel) at -10 mV (b), -25 mV (c) and -40 mV (d) are shown respectively. The piecewise function fitting graphs are introduced to better display the corner frequency (right panel), which will be discussed in #5. The lines are fitted results and the dots are experimental data.

At the same time, we do agree with this reviewer that, the difference in corner frequency should be more evident when an extreme state-of-charge (such as 10% and 90%) was examined, or in another redox system with significant lattice change during cycling. Unfortunately, when we tried to apply such extreme conditions in our study, the sample tended to rapidly lose activity (fading) during consecutive and long cycling under extreme potentials.

We have accordingly revised the manuscript to include the relevant results and

discussion on the influence of different offset potentials (Manuscript Page 7 and Supplementary Information Section 10).

3. **CRITICAL:** What was the electrolyte used? There are some slight mentions to 0.5 M KNO_3 but this is not clear (as the pH is critical in the analysis of PB structures and state of charge this must be reported in more detail – experimental details of the manuscript must be improved).

Response: We clarify that 0.5 mol/L KNO_3 was used as electrolyte throughout the work in the absence of pH buffer. KNO_3 not only served as electrolyte to reduce IR drop, but also provided sufficiently high concentration of K^+ for insertion/extraction. It was the mostly-frequently used electrolyte to study electrochemistry of Prussian Blue nanomaterials.

We appreciate this reviewer for pointing out the importance of pH. Therefore, we prepared 0.5 M KNO_3 + 50 mM $\text{KH}_2\text{PO}_4/\text{K}_2\text{HPO}_4$ buffer (pH 6.14) and compared the results in the presence and absence of buffer system. No obvious difference was observed in both cases (Fig. R5).

Fig. R5 oEIS of the same individual PBNP at 0.5 M KNO_3 (red) and 0.5 M KNO_3 + 50 mM $\text{KH}_2\text{PO}_4/\text{K}_2\text{HPO}_4$ buffer (blue). The lines are the fitted results, and the dots are experimental data.

We have accordingly strengthened the experimental section in the revised manuscript to better clarify some of the details (Manuscript Page 4 and Supplementary Information Section 2).

4. **CRITICAL:** In the absence of more experimental detail on the type of electrolyte and electrochemical measurements (including the presence or absence of redox probe in solution – which complies or not with Fig. 4) I am unable to evaluate the correctness of the equivalent circuit analysis conducted in section (theoretical analysis of oEIS) and equations employed therein.

Response: We apologize for the insufficient descriptions in the experimental section in the original submission, which has been strengthened in the revision. We clarify that the

electrochemical measurements were performed in the absence of additional redox probe in the solution. However, it was found that, the reduction of dissolved oxygen was responsible for the deviation of -90° phase at low frequency range as shown in Supplementary Fig. 21b. Besides, a 0.5 M KNO_3 solution was used as electrolyte throughout the work in the absence of pH buffer and redox probe (please refer to our response to question #3 and #7 for details).

The theoretical analysis conducted in this section “theoretical analysis of oEIS” is straightforward, but to assume that the behavior of the oEIS measurement can be modeled by a RC circuit, it requires more arguments on the meaning of, at least, the capacitance (in the electrochemical context of this paper).

Response: We have accordingly provided more discussion on the meaning of each element in the equivalent circuit as shown in Fig. 4a&b.

The regular Randles model in Fig. 4a was used to investigate the potential distribution between solution and electrical double layer as a function of frequency.

1) Resistor (R_s) described the solution resistance, which was determined to be ~ 42 Ohm. This value was consistent with literatures that used the same electrolyte and electrochemical cell design.

2) Double layer capacitor (C_{dl}) described the effect of electrical double layer, which was determined to be ~ 2.8 μF . The capacitance density was therefore 4.7 $\mu\text{F}/\text{cm}^2$, which is in good agreement with the previous values.

3) Polarization resistor (R_p) in series of warburg element (Z_w) were frequently used to describe the interfacial Faradaic reaction involving the reduction of dissolved oxygen in the present work (question #7 and Fig. R8 below). The Randles model was used to demonstrate one point, that the external voltage modulation nearly completely applied across the double-layer (and thus PBNP/electrode interface) within the frequency range below 100 Hz (Fig. 4c). It laid the foundation of the microscopic model to describe the oEIS of single PBNP as shown in Fig. 4b.

4) Nanoparticle resistor (R_{NP}) was used to describe a) the contact resistance at the nanoparticle-electrode junction, and b) charge transfer resistance used to describe electron/ion transport within the nanoparticle. We performed further control experiments to demonstrate this point. For example, the enhanced electrical contact (via vacuum drying), or the change in the offset potential, was able to monitor R_{NP} as expected. More descriptions were shown in Supplementary Information Section 10&11.

5) Nanoparticle capacitor (C_{NP}) described the charge storage capability of single PBNPs, including both the surface-limited charging layer and the diffusion-limited interior part. It was proportional to the volume of nanoparticles (Fig. 3g).

5. The key cut-off of the oEIS that defines the transition between diffusive and non-diffusive regimes at the frequency of 1.1 Hz are the core for this work. Can this cut-off be defined by plotting the optical amplitude as a function of the inverse of frequency and the inverse of the square root of the frequency in a unique superposed graphical for the readers taken their evaluation of this cut-off directly? The cut-off, as presented by the

authors, is graphically undefined in the sense that the readers cannot take it from the graphical by themselves. Can the authors provide more details on how this are taken methodologically speaking?

Response: According to this reviewer (and other reviewers') comment, we have improved the graphics by displaying both low and high frequency ranges in the same plot (Fig. R6 below, and Fig. 2e in the revised manuscript).

Once plotting the optical amplitude as a function of the inverse of the square root of the frequency ($f^{-0.5}$, as wisely suggested by this reviewer), it became clear that the curve was composed of two segments: a linear curve in the low frequency range (right part, $f^{-0.5}$), and a parabolic curve in the high frequency range (left part, f^{-1} or $(f^{-0.5})^2$). It was well consistent with the proposed mechanism. In order to unbiasedly determine the corner frequency, a piecewise function (Fig. R6 inset, in which f_{cutoff} is a parameter-to-be-fitted) was applied to fit the entire curve. For example, for the representative amplitude results shown in Fig. 2, the corner frequency was fitted to be 0.9 Hz (Fig. R6).

Fig. R6 (a) When plotting optical amplitude as a function of $f^{-0.5}$, the dependence can be well fitted by a piecewise function as shown in the (a) inset. (b) A part of (a) is enlarged to better display the transition between two trends at f_{cutoff} of 0.9 Hz. The red lines are the fitted results, and the blue dots are experimental data.

6. Rigorously, methods based on frequency analysis requires validations generally using Kramers-Kronig methods applied on the frequency-dependent data. Do the authors consider this method as an important criterion of validation of time/frequency-dependent data (presumably on the O^* function)? This would be important to validate the equilibrium condition for the charge state of the nanoparticles.

Response: We appreciate this valuable suggestion to verify the reliability of the frequency-dependent data. Kramers-Kronig methods were used to describe whether the systems satisfied the conditions of linearity, causality, stability, and finiteness. According to the Kramers-Kronig relations, the relevant equations were:

$$\Phi_{OTF}(\omega) = -\frac{\omega}{\pi} \int_0^{\infty} \frac{\ln(|OTF(x)|)}{x^2 - \omega^2} dx, \quad \ln(|OTF(x)|) = \frac{2}{\pi} \int_{-\infty}^{\infty} \frac{\Phi_{OTF}(\omega)}{x - \omega} dx.$$

And the simulation results were shown in Fig. R7.

According to this reviewer's comments, we have included the relevant results in the revised Supplementary Fig. 12.

Fig. R7 Bode plots of OTF amplitude (a) and phase (b). The red lines are Kramers-Kronig transform simulation results, and the blue dots are experimental data.

7. I agree with the equivalent circuit of (b) in Fig. 4 – but this requires clarification on the meaning of the capacitance. In other works, this equivalent circuit can only be assumed under specific charge load of nanoparticles. It also depends on certain electrolyte conditions (in the absence of redox probe, for instance). The equivalent circuit of Fig. 4(a) is not clear in which condition it was obtained (see my fourth comments above). There is a confusion here because the circuit of Fig. 4a only applies for the case where there is redox probe in solution (the traditional Randle circuit) and in which the Warburg element of the circuit represents the diffusion of ionic species. This circumstance is quite different from the non-diffusive case. Please, can the authors clarify? It is important to reconcile between diffusion and non-diffusive regime by addressing a correct meaning for the capacitance of the nanoparticles under the two regimes of reactions. The meaning of an electrochemical capacitance can be applied in both cases without the need of a Warburg element. Otherwise, there is a mixture of two analysis that are quite different in essence and that cannot be modelled as the authors did.

Response: We apologize for the insufficient descriptions in the experimental conditions that caused the confusion. We absolutely agree with this reviewer that the validity of as-proposed circuit was dependent on whether there were additional redox species in the solution or not.

We clarify that the electrochemical measurements were performed in the absence of additional redox probe in the solution. However, it was found that, the reduction of dissolved oxygen was responsible for the deviation of -90° phase at low frequency range as shown in Supplementary Fig. S21b. A polarization resistor (R_p) and Warburg element (Z_w) were introduced solely to have a better fitting to the conventional current-based EIS. As long as the oxygen was removed from the solution by purging Argon bubbles, the deviation was reduced (Fig. R8). Because the reduction of oxygen did not contribute any

optical signals, this could not affect our subsequent analysis.

Fig. R8 (a) The Nyquist plot of the overall electrochemical cell before (red) and after (blue) removing the dissolved oxygen. The corresponding Bode plots of amplitude (b) and phase (c) are also provided.

The meaning of both elements had been explained in detail in our response to question #4. We have accordingly strengthened the equivalent circuit model section in the revision to better clarify the details (Supplementary Information Section 15).

8. Only with the further clarifications demanded in my comment number 7 is that I will be able to adequately validate the analysis conducted by the authors in equations (1) to (6).

Response: We appreciate this reviewer for the valuable comments that have helped us to better clarify some of the experimental details. In the revision, we have conducted further experiments, and included corresponding results and discussion regarding optical transfer function, state-of-charge (offset potential), electrolyte (pH), redox species, and meanings of each element in the equivalent circuit. We have also improved the graphic to better display the transition (corner frequency f_{cutoff}). We sincerely hope the revision should have addressed this reviewer's concerns to judge the theoretical analysis in this work.

Reviewer #2 (Remarks to the Author):

The authors present an electro-optical imaging approach that measures the scattered light intensity of single Prussian blue particles as a function of an applied voltage waveform. One major advance of this study over previous single nanoparticle electro-optical studies (refs 22-24) is the electrochemical impedance waveform, which provides unique insight

into the ion insertion process. The scattering intensity versus frequency supports two different ion insertion kinetic regimes (i.e., mass transport-limited regime and surface kinetic-limited regime). This paper is a very nice piece of work and should be published in Nature Communications. However, it would be helpful to the community if the authors could clearly explain how the scattering signal quantitatively relates to charge inserted in the particles (see comments below).

Response: We very much appreciate this reviewer's enthusiasm and valuable comments that have helped us to significantly improve the manuscript. Detailed point-by-point responses are listed below.

1. Can the authors clarify what they mean by "an equilibrium potential of -25 mV was observed" in line 133.

Response: We clarify that the 'equilibrium potential' herein meant 'formal potential' where the state-of-charge was 50% PB–50% PW. In order to support this point, optical response of single PBNPs as a function of sweeping potential from -300 to +250 mV (scan rate: 5 mV/s) was provided (Fig. R3a above). It revealed a formal potential of -25 mV, and a quasi-linear dependence of optical intensity on potential during -55 and 5 mV. Note that different individuals exhibited different formal potentials that we would further describe in question #5 (Fig. R11 below).

In order to avoid miscommunications, the term 'equilibrium potential' was replaced by 'formal potential' in the revised manuscript. We have accordingly strengthened the formal potential section in the revision to better clarify the details (Supplementary Information Section 3).

2. In lines 130-134, the authors begin to imply that the scattering signal is immediately proportional to charge inserted in the particle. It might be useful to explain at this point that the scattering signal does not immediately report on the ion insertion content in the host. In line 181, the authors state "the optical intensity was quantitatively dependent on the oxidation state of the PBNPs". At this point in the manuscript, the authors have only demonstrated that the scattering signal changes in a rational way with the applied potential waveform. There is no quantitative relationship between charge inserted in the particle and the scattering signal (yet).

Response: We definitely agree with this reviewer that the original results were not sufficient to support the linear dependence of optical intensity with its ion content. In order to address this concern and to build a more quantitative correlation between optical intensity and ion content, we had further conducted a galvanostatic charging experiment. The charging current was 5×10^{-8} ampere due to the extremely low surface coverage of PBNPs (0.1%).

By recording the electrochemical current (to quantify charge or ion content) and optical scattering responses of single PBNPs simultaneously, we were able to quantitatively examine the dependence of scattering intensity with the ion content. Our

results demonstrated that the scattering intensity of single PBNPs linearly decreased with the increasing K^+ insertion content. There was a good linear relationship when the state-of-charge was around 25%–75%. Note that this method was adopted in our previous study to correlate the optical signal with state-of-charge of single $LiCoO_2$ nanoparticles (*J Am Chem Soc*, 2017, 139, 186).

According to this reviewer's comment, we have included relevant results in the revised Supplementary Information Section 8.

Fig. R9 Linear dependence of scattering intensity on the state-of-charge of single PBNPs as revealed by a galvanostatic charging experiment.

3. In line 193-194, the authors assume “the maximal optical amplitude at the plateau region was utilized to quantify the total charge that the particular PBNP could uptake”. A very recent Nature paper by Merryweather et al. also used a scattering method to study Li-ion insertion in $LiCoO_2$ particles. It is not straightforward to relate the scattering intensity to ion insertion content. Can the authors provide more justification for this important assumption in this work (Merryweather, A. J.; Schnedermann, C.; Jacquet, Q.; Grey, C. P.; Rao, A. Operando Optical Tracking of Single-Particle Ion Dynamics in Batteries. *Nature* 2021, 594 (7864), 522–528). Can the authors consider details in the SI of that Nature paper and explain how the proposed method in Section 6 of the SI is a reliable approach to quantitatively relate scattering intensity to charge inserted?

Response: We appreciate this reviewer for bringing this relevant and recent paper into our attention, which has been cited and briefly discussed in the revision.

We would like to kindly point out that, our work focused on the total scattering from a single PBNP that was smaller than the optical diffraction limit, while the reference work (*Nature* 2021, 594, 522) was superior to map the ion transport pathways by imaging the local variations of scattering from an irregular 10-micron sized $LiCoO_2$ particle.

There were a few reasons to ensure a more straightforward and quantitative relationship between optical signal and ion content in our study. First, because as-prepared PBNPs were around 100–300 nm and of regular cubic-shape, they appeared

as a round dot following two-dimensional Gaussian distribution. The integration of all pixels in the round pattern should have included nearly all photons collected by the objective. In contrast, for micron-sized particles with irregular morphology, it was extremely difficult, if not impossible, to predict the spatial distribution of scattered photons, particularly when considering the vertical dimension (thickness). Second, it was believed that the scattering of PBNPs at ~700 nm was due to the resonant Rayleigh scattering, because the incident wavelength was consistent with the absorption band of Fe-Fe intervalence charge transfer. Because absorption of nano-sized object was less sensitive to its morphology, it would be more reliable to reflect the ion content. Therefore, our results (Fig. R9) have clearly demonstrated the linear dependence between optical scattering intensity of single PBNPs and the ion content (state-of-charge).

According to this reviewer's comment, we have included relevant discussion in the revised Supplementary Information Section 8.

4. What do the authors mean by "corner frequency" in line 196?

Response: We clarify that the 'corner frequency' herein meant the transition frequency when single PBNP switched from surface-limited charging process to diffusion-limited ion insertion process. In order to better display the transition, we have improved the graphics by displaying both low and high frequency ranges in the same plot (Fig. R10 below, and Fig. 2e in the revised manuscript).

Once plotting the optical amplitude as a function of the inverse of the square root of the frequency ($f^{-0.5}$), it became clear that the curve was composed of two segments: a linear curve in the low frequency range (right part, $f^{-0.5}$), and a parabolic curve in the high frequency range (left part, f^{-1} or $(f^{-0.5})^2$). It was well consistent with the proposed mechanism. In order to unbiasedly determine the corner frequency, a piecewise function (Fig. R10 inset, in which f_{cutoff} is a parameter-to-be-fitted) was applied to fit the entire curve. For example, for the representative amplitude results shown in Fig. 2, the corner frequency was fitted to be 0.9 Hz (Fig. R10).

Fig. R10 (a) When plotting optical amplitude as a function of $f^{-0.5}$, the dependence can be well fitted by a piecewise function as shown in the (a) inset. (b) A part of (a) is enlarged to better display the transition between two trends at f_{cutoff} of 0.9 Hz. The red lines are the fitted results, and the blue dots are experimental data.

5. In line 213, the authors state they examined 30 PBNPs with “good activity”. There are more than 30 eligible objects in Fig 1b. Does this mean not all particles are active? If so, then the above assumption in comment 3 deserves more serious attention because the applied potential is applied to the entire electrode but the amount of charge injected into every particle is different. The “proportionality constant” approach described in Supplementary Section 6 does not explain the fraction of inactive particles.

Response: We agree with this reviewer that, although ~100 PBNPs existed in the wide-field dark-field image, only ~30 of them were chosen for further investigations. The reasons are as follows. First, with the present sample preparation procedures, ~20 individuals were found to be dimers, trimers and other kinds of aggregates (Fig. R11). Second, the formal potential of each individual had been examined by monitoring the optical intensity as a function of potential (please refer to our response to Question #2 from Reviewer 1). It was a consequence of structural and compositional heterogeneity during synthesis. We have accordingly selected the ones with formal potential in the range between -0.05 and 0 V, which accounted for the largest portion (Fig. R12 below). By applying these two criteria, it was found that, only ~30% individuals were applicable for further investigations.

Fig. R11 The dimers and aggregates in the wide-field dark-field image, scale bar: 100 nm.

Fig. R12 The distribution of formal potential of around 100 PBNPs, the inset is the representative formal potential of -25 mV.

6. In Fig. 2b, the authors could consider using mass transport-limited or diffusion-limited and surface-limited instead of diffusion-control and pseudocapacitance-control.

Response: We agree with this reviewer and have revised the manuscript accordingly.

Reviewer #3 (Remarks to the Author):

This is a very interesting work that applies the methodology of electrochemical impedance spectroscopy (EIS) at the single nanoparticle level. It is performed on PB Prussian Blue nanocubes. This choice of nanomaterial is relevant because it can be considered as a model of a solid-state charge storage nanodevice. While most charge storage activities of nanomaterials are studied at the level of a large ensemble, here the charge storage performance is studied at the level of a single particle, through the indirect study of the variation of its optical properties during EIS sollicitation. The amplitude and phase of the optical response of individual nanocubes to a sinusoidal potential waveform are then discussed and analyzed as would be the response to electrochemical current.

Various relevant and innovative information is obtained from this work. It totally fulfills the standards of Nature Communications. However, the following points should be discussed or reconsidered before publication.

Response: We very much appreciate this reviewer's comments that have helped us to improve the quality of our manuscript. Detailed point-by-point responses are provided below.

1. First of all, the methodology is sound and innovative. Indeed, works proposing an EIS analysis at the level of the individual NP are rare. It is also the first time that it is applied to the monitoring of individual nanoparticles by dark field optical microscopy. In this respect it paves the way to many other capacitance and impedance assessment at the single NP level. However, it seems that the concept of optical impedance spectroscopy has been quite developed and implemented in SPR microscopy by the late NJ Tao's group (even with the help of the lead author of this work). This should be recalled in the context and state of the art of this SPR. A comparison of the two optical EIS implementations or methodologies, e.g., in terms of sensitivities or temporal/frequency resolutions, could also be appreciated.

Response: We are grateful for this reviewer's acknowledgements on the pioneering study of optical impedance spectroscopy/microscopy proposed by Dr. NJ Tao and co-workers (including the lead author of this work indeed, *Nature Chemistry*, 2011, 3, 249, *Advance Materials* 2015, 27, 6213, *Ann Rev Anal Chem* 2017, 10, 183), which has been cited and described in the revised manuscript.

In SPR-based electrochemical impedance microscopy (p-EIM), the gold film acted as an optical-electrochemical conversion interface and exhibited a large background charging/discharging. Therefore, optical amplitude was largest for bare gold electrode, and the presence of object (such as cell, bacteria and nanomaterials) inhibited the background charging and decreased the optical amplitude. In other words, it was a 'turn-off' mode detection. The introduction of dark-field microscopy (DFM) in the present work enabled a 'turn-on' version of optical impedance imaging which is more suitable for studying single nanoparticles. In this work, optical amplitude directly came from the PBNPs themselves in a near-zero background.

We have accordingly revised the manuscript (Page 2&3) to include relevant discussion on the optical impedance microscopy.

2. The context of the study is related to pseudocapacitance. The authors recall the controversy central to this notion, not always accepted, where both double layer and faradaic behavior are observed. However, it is quite difficult to apprehend from the data presented whether the system response belongs or not to a pseudocapacitance behavior. 2.1 Indeed the EIS response is modelled by a serie RC circuit which may be an oversimplified description of a pseudocapacitor. Even ref 38 states that the RC serie description is too simple and a transmission line behavior would be more appropriate. It is indeed expected that the EIS response would show some charge transfer processes with diffusion and capacitive contributions. See for example the guidelines suggested by Gogotsi and Simon *Adv Energy Mater* 2019 1902007. It is then possible that the EIS data analysis is over interpreted as yielding pseudocapacitance behavior. Plotting the data in a Nyquist plot would show the EIS shows only a RC type semi-circle response without diffusive or even pure capacitive branches expected for the pseudocapacitor systems. One could then wonder if the system can be really named pseudocapacitor. It may be that the NP behavior is very different from the generally studied film electrodes.

Response: We appreciate this reviewer for bringing this relevant paper into our attention, which has been cited in the revision. In order to better illustrate the pseudocapacitive behavior, we have improved the graphics by displaying both low and high frequency ranges in the same plot (Fig. R13 below, and Fig. 2e in the revised manuscript).

Once plotting the optical amplitude as a function of the inverse of the square root of the frequency ($f^{-0.5}$), it became clear that the curve was composed of two segments: a linear curve in the low frequency range (right part, $f^{-0.5}$), and a parabolic curve in the high frequency range (left part, f^{-1} or $(f^{-0.5})^2$). It was well consistent with the proposed mechanism. In order to unbiasedly determine the corner frequency, a piecewise function (Fig. R13 inset, in which f_{cutoff} is a parameter-to-be-fitted) was applied to fit the entire curve. For example, for the representative amplitude results shown in Fig. 2, the corner frequency was fitted to be 0.9 Hz (Fig. R13).

Fig. R13 (a) When plotting optical amplitude as a function of $f^{-0.5}$, the dependence can be well fitted by a piecewise function as shown in the (a) inset. (b) A part of (a) is enlarged to better display the transition between two trends at f_{cutoff} of 0.9 Hz. The red lines are the fitted results, and the blue dots are experimental data.

Such transition provided vivid evidence to demonstrate the pseudocapacitive characteristics of single PBNPs. At the same time, the oEIS curve was converted to impedance displayed in Bode plot to indicate the applicability of equivalent circuit (Fig. R2 above, and Supplementary Information Section 7). Please refer to our response to Question #1 from Reviewer 1 for details.

2.2 One way to confirm the behavior would be to perform the same analysis at another state of charge of the NP, say another electrode potential, especially since there is a large potential range which can be explored without going to full charge/discharge of the material.

Response: We agree with this reviewer that it was important to examine the influence of electrode potential (state of charge) on the oEIS. According to this reviewer's comments, we had conducted further experiments at different offset potentials to examine the influence of state-of-charge on the OTF as well as corner frequency. It was found that,

while the maximal optical amplitude was indeed observed at the formal potential of PBNPs (-25 mV), the corner frequency and the depth of surface charging layer were more or less independent on the state-of-charge, at least in the range of formal potential ± 15 mV (corresponding to state-of-charge 30~70%). We attributed such stability as the same face-centered cubic crystal structure and the similar lattice parameters between Prussian Blue (oxidized form) and Prussian White (reduced form).

We have accordingly revised the manuscript to include the relevant results and discussion on the influence of different offset potentials (Manuscript Page 7 and Supplementary Information Section 10). Because relevant question has also been raised by Reviewer 1, please refer to our response to Question #2 from Reviewer 1 for details.

2.3. If the trend is not confirmed at other NP states of charge I wonder if the regimes presented in Fig 2a and 2b are correct. The pseudocapacitance region seems a pure capacitance behavior to me. The diffusion regime is rather a RC behavior while at lower frequency a pure resistive behavior is observed.

Response: We hope that our responses to previous questions 2.1 and 2.2 have addressed this reviewer's concern regarding this point.

2.4. Definitely, the conclusion relative to the thin depth of charging is likely correct. Indeed the apparent optical 'resistance' corresponds to a very small amount of charge/discharge when comparing the ca.80 optical units of the 25 mV amplitude pulse compared to the 2000 optical units required for full charge/discharge (Fig 1). As a comment, this ensure that the EIS measurement is indeed not perturbing much the overall composition of the NP, as requested for EIS experiments.

Response: We agree with this reviewer that a subtle disturbance was necessary to acquire meaningful EIS. Sensitive dependence of optical scattering intensity with the ion content, as well as the periodic modulation and Fourier transform, ensured the superior sensitivity of the present technique to resolve oEIS of single PBNPs under a voltage modulation amplitude of 20 mV.

3. The open circuit charge state of the NP is approximately 0.5. Is this value common or was it intentionally obtained during synthesis? This should be detailed.

Response: We clarify that the formal potential of single PBNPs under the present experimental conditions (0.5 M KNO_3 , ITO) exhibited heterogeneity among different individuals. The formal potential of each individual had been examined by monitoring the optical intensity as a function of potential (Fig. 1c). Different individual revealed different formal potential, which was a consequence of structural and compositional heterogeneity during synthesis as suggested by this reviewer. Our results exhibited that the formal potential of around -25 mV was dominated (Fig. R14 below). Therefore, we applied an offset potential of -25 mV and have intentionally selected the ones with formal potential in the range between -0.05 and 0 V for further investigations.

Fig. R14 The distribution of formal potential of around 100 PBNPs, the inset is the representative formal potential of -25 mV.

We have accordingly provided relevant details in Supplementary Information Section 3.

4. Actually it also means that the NP studied by EIS is half-charged and that the scheme in fig 2f presented to describe the PB charge/discharge is not correct. The calculation of the depth of surface charging may also be underestimated.

Response: We agree with this reviewer that, under our experimental conditions, PBNPs were half-charged. We have accordingly strengthened relevant discussion to clarify that PBNPs was composed of 50% PB cubic cells and 50% PW cubic cells at the formal potential. Our intention to use Fig. 2f is mainly to illustrate the general reaction process of PB to PW. It was believed that the different cubic cells were evenly mixed with each other from surface layer to interior parts. The subsequent charging/discharging still go through surface layer first and then interior part. Therefore, the estimation on the depth of surface charging layer remained fair. This point was supported by the experimental results that the corner frequency and the depth of surface charging layer were independent on the state-of-charge, at least in the range of formal potential ± 15 mV (corresponding to state-of-charge 30~70%, Fig. R4 above). It has been explained in our response to question 2.2.

5. The same authors have published other works related to the same PBNPs with dark field monitoring. One of their conclusions was that the NP activity depended strongly on the contact resistance to the electrode material. I wonder how they could circumvent this issue in the present work? I also wonder if finally this limitation is not the charge transfer resistance revealed by the EIS evaluation?

Response: We appreciate this reviewer for pointing out our previous work (*J Am Chem*

Soc, 2020, 142, 33, 14307), which investigated the influence of electrical contacts on the apparent activity of single nanoparticles. We clarify that this point remained valid, and the present oEIS offered a promising capability to quantify the contact resistance by analyzing R_{NP} .

As an example, we collected and compared the oEIS of a very same individual before and after drying the sample in a vacuum chamber (10^{-4} Pa) by 1 hr. As shown in Fig. R15 below, the value of R_{NP} was found to ~ 20 times lower after the vacuum drying. This result not only demonstrated the capability of oEIS for quantifying the contact resistance of single nanoparticles, but also provided a more feasible protocol to enhance the electrical contacts by vacuum drying (than our previous method of metal sputtering). Systematical and detailed results regarding this point are beyond the scope of this work, and will be published elsewhere.

Fig. R15 The normalized IOTFI of a very same individual PBNP before (red) and after (blue) vacuum drying (blue). The lines are the fitted results, and the dots are experimental data.

We have accordingly provided relevant details in Supplementary Information Section 11.

6. As a general comment, pseudocapacitors have been mostly studied by cyclic voltammetry, for example by varying the CV scan rate. This would be equivalent in some respects to the EIS measurement. Did the authors considered such studies which would more visually present the pseudocapacitor behavior?

Response: We clarify that similar trends can also be observed by applying a triangle waveform (as used in CV method), but the performance was compromised. According to this reviewer's comment, we have conducted further experiments. Representative optical intensity curves under CV (triangle waveform) and EIS (sinusoidal waveform) conditions are displayed and compared as shown in Fig. R16 below. It is clear that the optical response under CV condition significantly deviated from sinusoidal waveform, making the Fourier transform less reliable. This result is completely understandable because it is the whole point to use sinusoidal waveform (instead of triangle waveform) in EIS measurements.

Fig. R16 Representative scattering intensity curves when applying triangle waveform (a) or sinusoidal waveform (b) potential modulation of 0.01 Hz.

7. Finally, one may have shown the electrochemical behavior of large ensemble or film of NPs showing for example their electrochemical current based EIS rather than the oEIS. One could wonder if the oEIS behavior fits the electrochemical EIS one?

Response: According to this reviewer's comment, we have further collected the conventional (current-based) EIS of a large ensemble of PBNPs, as shown in Fig. R17 below. Both of two methods show the pseudocapacitive behavior of PBNPs.

A linear trend with a slope of $\sim 72^\circ$ is clearly observed in the Nyquist plot. It is between the feature of a pure-capacitor (90°) and that of a pure Warburg element (45°), demonstrating that the electrode process of PBNPs is not completely under diffusion control and it also exhibits good capacitive behavior. This result is also consistent with previous reports on conventional EIS of PBNPs (*RSC Adv.*, 2016, 6, 109340-109345).

Fig. R17 The Nyquist plot of ensemble PBNPs, the inset is the commonly used equivalent circuit diagram. The blue line is the fitted results, and the red dots are experimental data.

Comments on the manuscript NComm-21-37657

Title: **Determining the depth of surface charging layer of single Prussian Blue nanoparticles with pseudocapacitive behavior**

Authors: Ben Niu, Wengxuan Jiang, Mengqi Lv, Sa Wang, Wei Wang

The revised version of the manuscript was appropriately conducted by the authors. They appropriately addressed all the questions (some were answered with additional experimental data) that I had raised in such a way that my concerns were fulfilled. Accordingly, I am now supportive of the publication of this revised text. I would just suggest that the short and final section named as **Results** would be better addressed as **Conclusion**.

Reviewer #2 (Remarks to the Author):

The authors have satisfactorily answered my questions and addressed my comments. However, I still have one question about SI Section 8. The authors state "By recording the electrochemical current (to quantify charge or ion content) and optical scattering responses of single PBNPs simultaneously, we were able to quantitatively examine the dependence of scattering intensity with the ion content." To obtain the state of charge x-axis in Figure S13, the authors must assume the charge is distributed equally among a known mass/amount of particles on the substrate. What assumptions were made in this state of charge calculation?

Merryweather, A. J.; Schnedermann, C.; Jacquet, Q.; Grey, C. P.; Rao, A. Operando Optical Tracking of Single-Particle Ion Dynamics in Batteries. *Nature* 2021, 594 (7864), 522–528.
<https://doi.org/10.1038/s41586-021-03584-2>. showed that single particles undergo ion insertion processes at different rates and times during the galvanostatic charging experiment. The above assumptions could hold if all the PB trajectories show identical rates, onset time, and intensities (after correcting for particle size-dependent volume effects).

In summary, it would be good to discuss the assumptions in SI Section 8 and how those assumptions could impact the final conclusions.

Responses to Reviewers' Comments

Reviewer #2 (Remarks to the Author):

Comments on the manuscript NComm-21-37657A

Title: Determining the depth of surface charging layer of single Prussian Blue nanoparticles with pseudocapacitive behavior

The authors have satisfactorily answered my questions and addressed my comments. However, I still have one question about SI Section 8.

The authors state "By recording the electrochemical current (to quantify charge or ion content) and optical scattering responses of single PBNPs simultaneously, we were able to quantitatively examine the dependence of scattering intensity with the ion content." To obtain the state of charge x-axis in Figure S13, the authors must assume the charge is distributed equally among a known mass/amount of particles on the substrate. What assumptions were made in this state of charge calculation?

Merryweather, A. J.; Schnedermann, C.; Jacquet, Q.; Grey, C. P.; Rao, A. Operando Optical Tracking of Single-Particle Ion Dynamics in Batteries. *Nature* 2021, 594 (7864), 522–528. <https://doi.org/10.1038/s41586-021-03584-2>. showed that single particles undergo ion insertion processes at different rates and times during the galvanostatic charging experiment. The above assumptions could hold if all the PB trajectories show identical rates, onset time, and intensities (after correcting for particle size-dependent volume effects).

In summary, it would be good to discuss the assumptions in SI Section 8 and how those assumptions could impact the final conclusions.

Response: We very much appreciate this reviewer's valuable comments that have helped us to further improve the manuscript.

In order to convincingly demonstrate that the overall scattering signal of single PBNPs smaller than optical diffraction limit was quantitatively dependent on its state-of-charge, we have performed additional experiment by simultaneously recording the faradaic current and optical traces of one PBNP during its collision. The apparatus and methodology were adopted from our previous publication (collision and oxidation of single LiCoO₂ nanoparticles studied by correlated optical imaging and electrochemical recording, *Anal. Chem.*, 2017, 89, 6050). While LiCoO₂ nanoparticles were studied in previous study, herein PBNPs was used instead. Briefly, a 50x50 μm² microelectrode was fabricated to reduce the background current. When applying a constant reduction potential (-300 mV) onto the electrode and allowing single freely-moving PBNPs in the suspension to stochastically collide onto the electrode, a transient reduction current was recorded after the collision-and-stay of single PBNPs. By doing so, it was ensured that the electrode current was solely contributed by the particular nanoparticle. Since early 2000s, it has been a very powerful strategy pioneered by Lemay, Bard, Compton, and many others

(Acc. Chem. Res. 2016, 49, 2625), which was known as single nanoparticle collision/impact electrochemistry. Our contribution was to employ an optical microscopy to simultaneously record the entire collision-and-reaction process, and to quantitatively compare the optical signals with electrochemical current.

Fig. R1 (a) Schematic illustration of single PBNP collision events with correlated surface plasmon resonance microscopy (SPRM) imaging and electrochemical recording. (b) The bright-field (top panel) and plasmonic images (bottom panel) of Au microelectrode. (c) Correlation between the state-of-charge/electric quantity and the SPRM intensity for the PBNP. Transient SPRM intensity curve (d) and sequential electrochemical current (e) of a single PBNP, scale bar: 5 μm .

According to this reviewer's comment, we have conducted further experiments on single PBNPs collision. As shown in Fig. R1d, at 0.91 second, the nanoparticle collided on the electrode and therefore led to a sudden increase in the optical signal. Then, electrochemical reduction of the nanoparticle gradually increased the optical signal, indicating the gradual conversion from PB to PW. This point was confirmed by the simultaneously recorded reduction current (Fig. R1e). In this experiment, it was ensured that the optical signal and electrode current was from the same individual PBNP (please refer to Anal. Chem. 2017, 89, 6050 for details). If we plotted the optical intensity as a function of quantity of injected electrons (integration of current), there was a quasi-linear dependence of optical scattering signal on the state-of-charge, especially in the range between 40–80% (Fig. R1c).

It was necessary to clarify that, surface plasmon resonance microscopy (SPRM) rather than dark-field microscopy (DFM) was employed to obtain the results shown in Fig. R1a. It was because a micron-sized electrode was required for this experiment to suppress the background current. The edge of microelectrode resulted in a rather high optical background in DFM. However, since both SPRM and DFM measured the optical scattering signal, we believed the monotonic dependence of optical scattering on

state-of-charge remained valid in both cases.

We agree with this reviewer that significant spatiotemporal heterogeneity indeed existed. When charging single nanoparticles, as clearly revealed in the recent study (Merryweather, A. J.; Schnedermann, C.; Jacquet, Q.; Grey, C. P.; Rao, A. Operando Optical Tracking of Single-Particle Ion Dynamics in Batteries. *Nature* 2021, 594 (7864), 522–528), we would like to believe that the overall scattering intensity was able to quantitatively report the state-of-charge of single nanoparticle as long as it was smaller than the optical diffraction limit (~300 nm). First, our results in single nanoparticle collision electrochemistry have clearly supported this point for PBNPs. Second, nearly all individual PBNPs we investigated displayed a monotonic and smooth intensity curve during electrochemical charging/discharging cycles. In addition, in our previous study on single LiCoO₂ nanoparticles (~200 nm size), monotonic dependence was also observed (*Anal. Chem.* 2017, 89, 6050).

We would like to kindly point out that, our work focused on the total scattering from a single PBNP that was smaller than the optical diffraction limit, while the reference work (*Nature* 2021, 594, 522) was superior to map the ion transport pathways with sub-particle spatial resolution by imaging the local variations of scattering from an irregular 10-micron sized LiCoO₂ particle. There were a few reasons to ensure a more straightforward and quantitative relationship between the optical signal and the ion insertion content in our study. First, because as-prepared PBNPs were around 100–300 nm and of regular cubic-shape, the diffraction effect allowed for accessing the overall change in morphology and refractive index. This scenario was in contrast to micro-particles with a size of tens of microns and irregular morphology. Second, it was believed that the scattering of PBNPs at ~700 nm (wavelength) was due to the resonant Rayleigh scattering, because the incident wavelength was consistent with the absorption band of Fe-Fe intervalence charge transfer. Since absorption of nano-sized object was less sensitive to its morphology, it would be more reliable to reflect the overall ion content within the entire nanoparticle.

In summary, we have provided experimental results and relevant discussion to demonstrate that, the linear dependence between optical scattering intensity of single PBNPs and its state-of-charge was reliable under these two conditions: 1) nanoparticle size was smaller than optical diffraction limit, and 2) the state-of-charge was close to 50% in our experiment.

According to this reviewer's comment, we have included relevant results in the revised Supplementary Information Section 8.

REVIEWERS' COMMENTS

Reviewer #2 (Remarks to the Author):

Accept.